# Development and Analysis of an Intensified Batch-Fed Wine Fermentation Process

**Konrad V. Miller** [1,2,*], **Even Arefaine** [1], **Ardic Arikal** [1], **Annegret Cantu** [2], **Raul Cauduro Girardello** [2], **Anita Oberholster** [2], **Hildegarde Heymann** [2] and **David E. Block** [1,2]

1   Department of Chemical Engineering, University of California, One Shields Avenue, Davis, CA 95616, USA; agarefaine@ucdavis.edu (E.A.); oaarikal@ucdavis.edu (A.A.); deblock@ucdavis.edu (D.E.B.)
2   Department of Viticulture and Enology, University of California, One Shields Avenue, Davis, CA 95616, USA; acantu@ucdavis.edu (A.C.); rgirardello@ucdavis.edu (R.C.G.); aoberholster@ucdavis.edu (A.O.); hheymann@ucdavis.edu (H.H.)
*   Correspondence: vonmiller@ucdavis.edu

**Abstract:** White wine fermentations are typically performed in an entirely batchwise manner, with yeast nutrients only added at the beginning of fermentation. This leads to slow (2+ weeks) fermentation cycle times, with large capital expenditures required to increase winery processing capacity. Prior attempts to speed fermentations via increasing temperature have resulted in unpalatable wine, and continuous fermentation processing is uneconomical and impractical in the winery setting. In this work, we measured yeast nutrient consumption as a function of fermentation progression at the 300 mL scale, and from this derived an equation to optimize yeast nutrient concentration as a function of fermentation progression. These findings were applied at the pilot scale in 150 L fermentors, which resulted in a 60% cycle time reduction versus "best practices" control fermentations. The resultant wines were compared via GC-MS as well as by a trained sensory panel. Organoleptic analysis found statistically significant, but overall, small differences in sensory characteristics between the control and process intensified wines. This intensified fermentation process shows great promise for fermented beverage producers wishing to maximize equipment utilization and debottleneck wineries or other beverage fermentation facilities.

**Keywords:** fermentation; process intensification; sensory; wine; bioprocess

## 1. Introduction

Wine production is the oldest bioprocess in existence, and has been a major driver in the advancement of biology, chemistry, and process science: pasteurization, for example, arose from Pasteur's efforts to prolong wine shelf life [1]. Despite this, wine fermentations are still quite primitive from a bioprocess engineering point of view. White wine fermentations are carried out by expressing juice from white grapes and adding an initial inoculum of cultivated yeast (*Saccharomyces cerevisiae*). At the most, industrial winemakers might measure and correct initial nutrient levels and control fermentation temperature via the use of a tank jacket or external heat exchanger. White wine fermentations are typically carried out at low (14–20 °C) temperatures [2]. These low temperatures result in slow process cycle times for a single batch of white wine, on the order of 1–3 weeks. Wine presents another challenge in its seasonality: wineries only operate during the "harvest" or "crush" period, September–November in the Northern Hemisphere, February–April in the Southern. Slow fermentations with only ~1/3 yearly uptime means that wineries are capital intensive versus breweries or distilleries with similar yearly throughputs.

Wine juice is a nitrogen limited media where *Saccharomyces* rapidly dominates [3], even when not inoculated. Yeast derives nitrogen from ammonia and alpha-amino acids, typically referred to as Yeast Assimilable Nitrogen (YAN) [4,5], with uptake following Monod style saturation kinetics. YAN is typically present in grape juice in the 100~300 mg/L level

(though lower and higher levels do occur naturally) and it is often corrected up to 350 mg/L by the addition of ammonia or amino acids by the winemaker prior to fermentation, or occasionally at the beginning of the exponential phase of yeast growth. Yeast consumes hexoses (in grape juice, glucose and fructose) in a non-growth associated manner and produce ethanol in proportion to hexose consumption. Metabolic activity is a strong function of temperature [6]; biomass deactivation is a function both of ethanol concentration and temperature. While Monod-style models have historically been utilized to understand wine fermentation kinetics, the application of genome scale metabolic models to explore the production of dilute sensory metabolites for fermentation optimization has also been performed [7].

The basic nature of industrial wine fermentations, coupled with the large global demand for wine, the capital-intensive nature of wineries, and recent advancements in fermentation engineering, make wineries a ripe target for fermentation process intensification. Process intensification is the science of increasing the productivity of existing processes and has been applied widely to industrial bioprocesses [8], such as pharmaceuticals [9] or biodiesel production [10,11]. Typical fermentation process intensification approaches include the engineering of optimized organisms, the application of fermentation engineering to improve process conditions, and the switch from totally batch operation to either fed-batch or continuous operation.

Increased first-principle understanding of wine fermentations has led to multiple attempts to accelerate wine fermentations. Increasing fermentation temperature with thermotolerant yeast [12] has led to faster fermentations at the cost of a significant negative sensory impacts. Attempts to convert wine fermentations to continuous processes [13] have similarly met with mixed success; while feasible at the benchtop scale, continuous fermentations have sterility requirements well outside the scope of current winery processing capabilities and are difficult and expensive to implement. For an intensified process to be widely applicable, it must readily slot into the processes and equipment of existing wineries while offering minimal sensory impact. It is also important to distinguish attempts to intensify other non-fermentative aspects of winemaking (for example, optimization of phenolic extraction, [14]) from fermentation process intensification.

Fed-batch (or semi-batch) fermentations represent a middle ground between totally batch fermentations, where an initial charge is fed to a tank and then reacted to completion; and continuous fermentations, where media is constantly added and removed from the fermentor. In a fed-batch fermentation, the nutrient is metered in over the course of the fermentation to maximize the tank's volumetric productivity. In this work, we designed a fed-batch fermentation process for wine fermentations, where YAN was metered in throughout the fermentation such that biomass growth is maximized, while leaving no more residual YAN at the end of fermentation than a typical fermentation. YAN consumption rates over the course of fermentations were measured at the 300 mL flask scale, with the fed-batch fermentations taking place at the 150 L pilot scale to demonstrate industrial feasibility. The wine was subjected to chemical, sensory, and safety analyses post bottling.

## 2. Materials and Methods

### 2.1. Flask Fermentations

To determine YAN uptake rate over the course of fermentation, flask fermentations were performed in triplicate in 500 mL Erlenmeyer flasks charged with 300 mL must at room temperature (22–23 °C). Frozen 2012 UC Davis Chardonnay juice was used as the fermentation medium, 24.4 Brix, 310 mg/L YAN, and pH 3.45. Fermentations were run in triplicate in a two-factorial experiment: low YAN (310 mg/L) or high YAN (1100 mg/L); low biomass (0.25 g/L) or high biomass (10 g/L). YAN was adjusted with Fermaid K (Scott Labs, Petaluma, CA, USA), fermentations were inoculated with *Saccharomyces cerevisiae* strain EC1118 (Scott Labs, Petaluma, CA, USA).

Brix, optical density (OD), and YAN were measured every 12 h until the end of fermentation, defined as no change in Brix over two concurrent measurements. Total and

viable cell counts were performed at the end of fermentation. Brix was measured via a DMA-35 handheld densitometer (Anton Paar GmbH, Graz, Austria). Optical density (OD) was measured in a Genesys 10S UV-VIS spectrophotometer (Thermo Scientific, Waltham, MA, USA) at 600 nm via disposable cuvette, with samples diluted to the 0.2–0.4 OD linear range for biomass measurement, blanked against similarly diluted unfermented juice [15]. YAN concentrations were determined in the UC Davis Pilot Winery Lab in Davis, CA as the sum of free amino nitrogen and ammonia ion concentrations in the fermentation juice. Free amino nitrogen (K-PANOPA kit, Megazyme, Wicklow, Ireland) and ammonium ion (Ammonia-Rapid kit, Megazyme, Wicklow, Ireland) kits were purchased from BSG Wines, Napa, CA, USA for use in a Gallery Discrete Analyzer (Thermo Scientific, Waltham, MA, USA). The samples were collected by filtering the cell suspensions with a 0.45 μm PVDF Millex-HV Syringe Membrane filter (Burlington, MA, USA) prior to YAN analysis. Samples were stored at −20 °C until YAN analysis. Cell suspensions were diluted to appropriate concentrations and studied using a Bright-Line hemacytometer (Hausser Scientific, Horsham, PA, USA) under a Zeiss light microscope at $400\times$ magnification to estimate total and viable cell concentrations. Cell slurries (20 μL) were added to 20 μL of methylene blue solution (0.4% methylene blue, 10% ethanol [95%], 0.4 M $KH_2PO_4$) and this mixture was contacted for 1 min before analysis. Blue cells were counted as dead while the cells with no color were counted as live.

Microsoft Excel was used to analyze flask fermentation data on a Windows 10 PC. YAN consumption rate per degree brix drop per unit OD, normalized to current YAN concentration, was expressed as a function of fermentation Brix and an exponential fit was applied. This fit was then integrated to estimate the maximum YAN consumption for the remainder of the fermentation as a function of current YAN, current Brix, and current biomass OD.

*2.2. Pilot Fermentations*

After the flask fermentations were analyzed, pilot-scale fermentations were performed in duplicate in stainless steel jacketed cylindrical tanks, with 150 L juice charged per tank, in the UC Davis Research Winery. Chardonnay juice was again used as the fermentation media, specifically 2019 UC Davis Chardonnay, 24.2 Brix, pH 3.63, 280 mg/L YAN.

Three types of fermentations were performed: a control (no nutrient addition), a semi-batch fermentation (Fermaid K fed twice daily to adjust YAN over the course of fermentation), and a recycle-semi-batch fermentation (same as the semi-batch, but inoculated with the harvested biomass of a previous fermentation for a high initial inoculum). All fermentations were carried out at a 16C jacket set point, with a direct inoculation of 0.25 g/L EC1118 yeast. For the recycle semi-batch, to mimic yeast recycle from a cone-bottom tank, 15 L juice from the two recycle-semi-batch tanks was removed and inoculated at 3.75 g EC1118 (i.e., inoculum of a 150 L fermentation at 0.25 g/L) and fermented to dryness in a 16C temperature-controlled room. The resulting wine was then agitated with a standing mixer and pumped back into the fermentors to serve as inoculum. The control fermentation's YAN was not adjusted, as 280 mg/L is considered as more than sufficient YAN for a healthy fermentation (Boulton 1996).

YAN, brix, and OD were measured twice per day until end of fermentation, as specified in the *Flask Fermentations* section. Total Fermaid K additions averaged 578 g per tank into the semi-batch runs, and 1024 g into the recycle-semi-batch runs. These data were measured at the time of sampling and used to determine the new YAN set point for the semi-batch and recycle-semi-batch fermentations, with YAN adjusted up to the level recommended by Equation (1). In this way, the flask fermentation experiments informed the rate of nitrogen addition as a function of fermentation progression. At the end of fermentation, the wines underwent solids settling in a −1 °C cold room and were then bottled under nitrogen blanketing. Finished samples were sent to ETS Labs (St. Helena, CA, USA) for ethyl carbamate analyses via HPLC-MS-MS (QQQ) to ensure the safety of the high-YAN

wines. Residual sugars of the finished wines were measured by the UC Davis Research Winery via Gallery Discrete Analyzer (Thermo Scientific, Waltham, MA, USA).

### 2.3. Wine Volatile Chemical Analysis

Wine volatile compounds were analyzed as described by Girardello et al. [16]. In summary, samples from each fermentation replicate were prepared and analyzed in triplicate within a month of wine sensorial analyses as follows.

A total of 3 g sodium chloride (NaCl) (Sigma-Aldrich, St Louis, MO, USA) was placed into 20 mL amber vials, and then 10 mL of wine were transferred into the vials. Moreover, 50 μL of a solution of 2-undecanone (10 mg/L prepared in 100% ethanol) was added to each vial as an internal standard. An automated headspace-solid phase microextraction-gas chromatography-mass spectrometry (HS-SPME-GC-MS) model 7890A gas controlled by Maestro (version 1.2.3.1, Gerstel Inc., Linthicum, MD, USA) was used to analyze the wines. Samples were exposed to a 1 cm polydimethylsiloxane/divinylbenzene/carboxen (PDMS/DVB/CAR) (Supelco Analytical, Bellefonte, PA, USA), 23-gauge solid-phase microextraction (SPME) fiber for 45 min. Helium was used as carrier gas at a flow of 0.8636 mL/min, in a DB-Wax 231 ETR capillary column (30 m, 0.25 mm, 0.25 μm film thickness) (J&W Scientific, Folsom, CA, USA) column. The method used was retention time-locked for the internal standard 2-undecanone at a constant pressure of 6.69 psi to prevent retention time drifting. Samples were agitated and warmed to 30 °C for five minutes, following the fiber introduction into the vial headspace in order to adsorb volatile compounds for 45 min. Then, the SPME fiber was desorbed into the column in split mode (10:1 split ratio) and the oven temperature was kept 40 °C for five minutes, increased to 180 °C at 3 °C/min, then heated to 250 °C at 30 °C/min with the total run time of 61.67 min. The samples were measured using synchronous scan and selected ion monitoring (SIM mode), the range of which was from 40 $m/z$ to 300 $m/z$.

Data were analyzed by MassHunter Qualitative Analysis software Version B.07.00 (Agilent Technologies). Samples were analyzed semi-quantitatively by normalizing each volatile compound peak area with the peak of 2-undecanone as internal standards. Volatile compounds were identified by mass spectrometry (MS) spectrum of the peaks found at the determined retention times and each peak spectrum was compared to the National Institute of Standards and Technology (NIST) (https://www.nist.gov, accessed on 1 July 2020) database for further confirmation. Statistical differences among the wines were analyzed using univariate analysis of variance (ANOVA) and the means were determined by Fisher's least significant differences (LSD). Principal component analysis (PCA) was performed to compare and visualize the relations between wines and volatile compounds. All statistical analyses were performed in XLSTAT (Microsoft Office Professional Plus 2010, version 14.0.7194.5000, Redmond, WA, USA).

### 2.4. Sensory Analysis

The sensory profile (aroma, taste, and mouthfeel) of six (two control, two semi-batch, and two recycle-semi-batch) US Chardonnay wines made in the University of California, Davis research winery was analyzed after 3 months of bottling. The descriptive sensory analyses (DA) were performed in the J. Lohr Wine Sensory Room at UC Davis, in March 2020, following methods outlined in Heymann [17]. For the panel, 13 judges (7 females and 6 males) were recruited from students, staff, and friends of UC Davis based on availability and interest. This project was approved by the UC Davis Institutional Review Board (IRB) with the IRB project number 1546503-1.

The panel participants were trained in four 60-min group training sessions, where the panelists generated and discussed sensory attributes by consensus. The sensory terms were anchored by corresponding reference standards made from typical household and grocery items to describe the wines. During the training, the panelists established a standard tasting procedure and familiarized themselves with the use of the data collecting software. All wines were served at room temperature in black glasses covered with plastic lids. Each

sample contained a constant volume of 40 mL of wine. Wine glasses were coded with randomized three-digit numbers that differed for each wine judge. The wines were served during testing across judges in a balanced and randomized Latin square order. They were rated in individual, ventilated, and light isolating tasting booths under white light. All wines were rated for each generated sensory term intensity in quadruplicate on an unstructured 15-cm line scale anchored by the wording "not present" to "very intense", except for viscous, where the wines were rated from "watery" to "very viscous". FIZZ network (version 2.47B, Biosystèmes, Courtenon, France) was used for data collection. Judges were required to expectorate the wine samples and wait thirty seconds between samples to clean their palates with water and unsalted crackers. Six samples were evaluated in one session with a 1-min break after three samples. At the end of each session, judges were given snacks, and at the completion of the study, they were compensated with a gift card.

A one-way MANOVA (Multi Analysis of Variance) for the main factor wine was calculated to check for overall differences among wines [17]. Furthermore, a three-way ANOVA (Analysis of Variance) with the factors wine, judge, and replicate as well as their corresponding two-way interactions was used to detect significant different attributes among wines. In those cases, where the effect of the wine was significant, but one of the interaction terms, including wine, a pseudo mixed model [18] was applied. Here, a new F-value was calculated with the mean sum of squares from the significant interaction as an error term for the factor wine/FPI treatment. Fisher's LSD (Least Significant Difference) was then performed on FPI treatment to find significant groupings. The significance level for all statistical tests was set to $p < 0.05$. All statistical tests were calculated using R-3.2.2.

## 3. Results

### 3.1. Flask Fermentations

Flask fermentations followed a typical batch kinetic profile: early growth of biomass in the first few days, followed by the consumption of hexoses until the end of fermentation. As expected, higher initial YAN resulted in higher biomass content and faster fermentations; higher initial biomass resulted in faster YAN consumption and faster fermentations, as seen in Table 1. The specific growth rate, $\mu\_max$, is in line with observations by Cramer and Coleman. As expected, higher YAN results in higher $\mu\_max$, as YAN is the growth-limiting nutrient in the fermentations. The fastest growth rate is observed at High YAN and Low Biomass, giving the most YAN per unit biomass and perforce the highest specific growth rate.

**Table 1.** Flask fermentation kinetics.

| Fermentation | Max Rate, °Brix/h | $\mu\_max$, 1/h | Fermentation Time, h |
|---|---|---|---|
| Control | 0.41 | 0.12 | 132 |
| High YAN, Low Biomass | 0.43 | 0.15 | 120 |
| Low YAN, High Biomass | 0.78 | 0.10 | 84 |
| High YAN, High Biomass | 0.89 | 0.10 | 84 |

The fermentation data were normalized for Brix consumption instead of fermentation time to account for varying fermentation temperatures. Hotter fermentations are faster, while cooler fermentations are slower. By plotting the YAN consumption rate per unit Brix drop, normalized (divided) by current optical density and current YAN (*Y*-axis), as a function of current fermentation Brix (*X*-axis), Figure 1 is derived. Although all four flask fermentations had vastly different conditions, the normalized YAN uptake rates changed together as a function of fermentation progression. This figure shows how over the course of a wine fermentation, the yeast biomass consumes less YAN on a per cell basis, indicating

the switch from primary to secondary metabolism, as well as the deactivation of yeast cells due to the presence of ethanol.

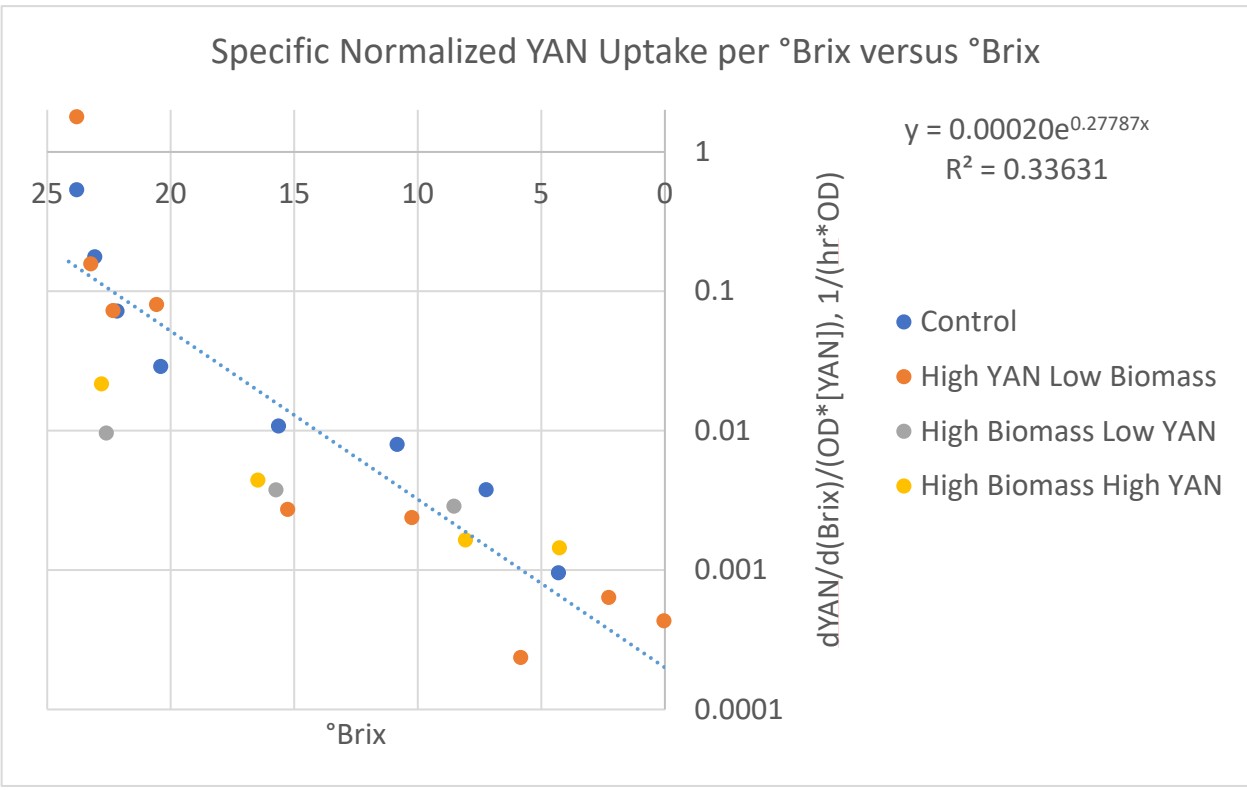

**Figure 1.** Consumption of YAN as a function of fermentation progression, current optical density, and current YAN. Data points captured 2x/day over the course of flask fermentations.

Integrating the resultant exponential fit results in Equation (1), which estimates the maximum YAN that the current biomass concentration can consume before the end of fermentation with only 30 mg/L YAN residual:

$$N_{Now} = e^{\left\{3.4 - \frac{OD*0.0002}{0.27787}\left[1 - e^{(0.22787 \cdot {}^{\circ}Brix_{now})}\right]\right\}} \tag{1}$$

where OD = optical density, $N_{Now}$ = maximum YAN the biomass can consume at current point in fermentation, and $Brix_{now}$ = current Brix. This equation can be applied to determine the maximum amount of YAN that the current biomass concentration can consume by the end of fermentation. By subtracting the current YAN level from $N_{Now}$, the maximum useful addition of YAN can be determined—any higher addition risks going unconsumed.

### 3.2. Pilot Fermentations

The intensified fermentations saw a substantial impacts versus the control, with substantial fermentation velocity, cell density, and overall cycle time improvements, as seen in Table 2 and Figure 2. The semi-batch fermentation had the highest specific growth rate, an expected result as it had a low initial inoculum compared to the recycle semi-batch, but much higher available YAN versus the control. The recycle-semi-batch had the fastest peak fermentation rate and reached 0 Brix before the other fermentations. The recycle semi-batch-fermentation had the lowest residual sugar at the end of fermentation, at half the value of the semi-batch fermentation. Interestingly, the control fermentation had an order of magnitude higher residual sugars at the end of fermentation. The control sample did have slightly lower overall YAN at the end of fermentation.

**Table 2.** Pilot fermentation kinetics.

| Trial | μ_max, 1/h | Max °Brix Rate, °B/h | Time to 0 °B, h | Residual Sugars at End, g/L | YAN at End (mg/L) | End Cell Number, cell/mL | End Cell Viability, % | Final Dry Cell Weight g/L |
|---|---|---|---|---|---|---|---|---|
| Control | 0.13 | 0.21 | 249.3 | 0.86 | 12.5 | $7.14 \times 10^7$ | 60% | 2.31 |
| Semi-Batch | 0.26 | 0.35 | 142.8 | 0.045 | 28 | $1.30 \times 10^8$ | 76% | 4.86 |
| Recycle Semi-Batch | 0.09 | 0.38 | 99.2 | 0.02 | 26.5 | $1.19 \times 10^8$ | 53% | 5.18 |

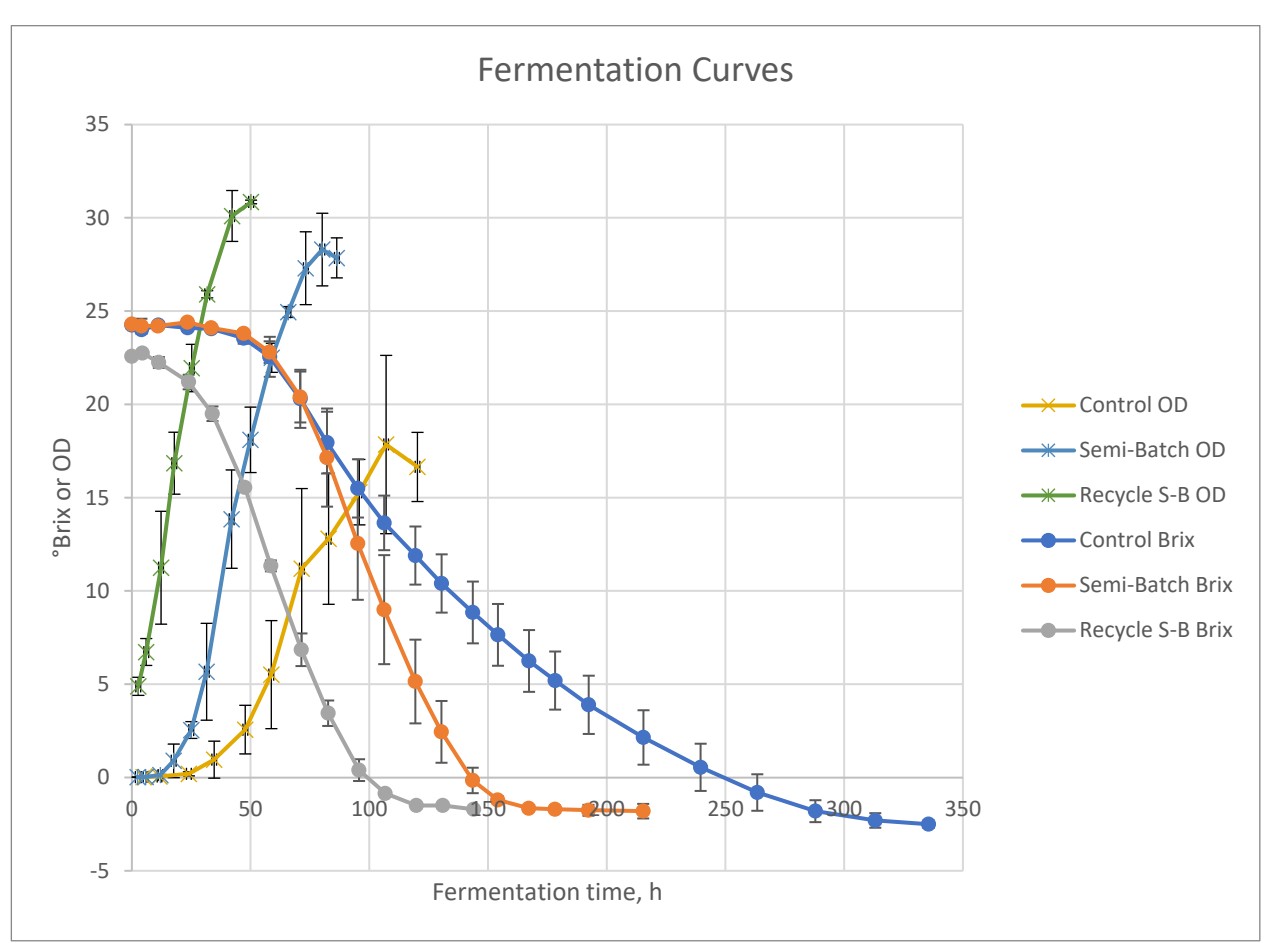

**Figure 2.** Fermentation curves showing Brix and Optical Density of the control, Semi-Batch (S-B), and Recycle-Semi-Batch (Recycle S-B) fermentations. The 95% confidence intervals are shown. The control slowly reaches a lower peak biomass compared to the intensified processes. The recycle-semi-batch starts at a much higher initial optical density due to the recycled inoculum. Substantial fermentation cycle time improvements are gained via the intensified processes.

Cell density and cell viability varied between conditions in interested ways. Both the semi-batch and recycle-semi-batch fermentation had more than double the end point dry cell weight of the control fermentation. Interestingly, the semi-batch fermentation had the healthiest cells at the end of fermentation, with 76% cell viability, as well as the highest cell number. The recycle-semi-batch had the lowest viability, an expected result given the additional age of the culture due to recycle.

All fermentations measured <30 mg/L residual YAN at the end of fermentation, and all fermentations were measured below ethyl carbamate detection threshold (>3 micrograms/mL) via HPLC MS/MS (QQQ) from ETS Labs.

### 3.3. Wine Volatile Chemistry Analysis

Table 3 shows the volatile compounds analyzed in the wines by GC-MS. From a total of 24 compounds found, 7 of them (29%) were shown to be significantly different among the samples. Recycle Semi-Batch had significantly lower levels of ethyl decanoate, ethyl octanoate, nerol, and 2-phenylmethyl alcohol and higher levels of isobutanol when compared to Control wines. Similarly to Recycle Semi-Batch, Semi-Batch also had lower levels of decanoate, ethyl octanoate, nerol, and 2-phenylmethyl alcohol than Control wines, although Semi-Batch and Recycle Semi-Batch did not show significant differences between each other regarding these compounds (except for ethyl decanoate, which the levels were the lowest in Recycle Semi-Batch). Damascenone levels were significantly higher in Semi-Batch when compared to Control and Recycle Semi-Batch (which were not significantly different from each other). Finally, nerolidol levels were found to be the lowest in Control wines when compared to Semi-Batch and Recycle Semi-Batch.

**Table 3.** Wine volatile compound levels and GC-MS parameters analyzed. Values are expressed as relative peak area multiplied by 1000. Bold compounds = significant different among the wines.

| Compound | Wine | | | GC-MS Parameters | | |
|---|---|---|---|---|---|---|
| | Control | Semi-Batch | Recycle Semi-Batch | Peak Number | RT [a] (min) | SIM Ions [b] |
| Ethyl acetate | 1722.62 ± 552.22 [a] | 1772.7 ± 6.32 [a] | 1246.24 ± 762.8 [a] | 1 | 3.02 | 43, 61, 88 |
| Ethyl isobutyrate | 4.41 ± 0.42 [a] | 7.11 ± 0.96 [a] | 12.62 ± 9.36 [a] | 2 | 4.39 | 43, 71, 86, 116 |
| Ethyl butyrate | 112.89 ± 39.34 [a] | 91.94 ± 3.59 [a] | 69.8 ± 49.72 [a] | 3 | 6.43 | 71, 88, 116 |
| Ethyl 2-metylbutyrate | 2.46 ± 0.29 [a] | 1.51 ± 0.00 [a] | 2.22 ± 1.52 [a] | 4 | 6.96 | 57, 102, 130 |
| Ethyl isovalerate | 2.96 ± 0.36 [a] | 4.04 ± 0.65 [a] | 4.61 ± 3.07 [a] | 5 | 7.57 | 85, 88, 130 |
| **Isobutanol** | 96.88 ± 1.51 [b] | 99.28 ± 0.03 [b] | 121.96 ± 6.4 [a] | 6 | 8.48 | 43, 55, 74 |
| Isoamyl acetate | 1300.75 ± 596.43 [a] | 1249.05 ± 152.01 [a] | 1011.41 ± 766.7 [a] | 7 | 9.84 | 55, 87, 130 |
| β-Myrcene | 1.58 ± 0.09 [a] | 1.66 ± 0.04 [a] | 2.05 ± 0.23 [a] | 8 | 11.63 | 69, 93, 136 |
| α-Terpinene | 0.25 ± 0.03 [a] | 0.32 ± 0.03 [a] | 0.35 ± 0.03 [a] | 9 | 12.05 | 93, 121, 136 |
| Limonene | 9.89 ± 0.73 [a] | 10.00 ± 0.05 [a] | 10.77 ± 0.08 [a] | 10 | 12.92 | 68, 93, 136 |
| Isoamyl alcohol | 3178.77 ± 154.86 [a] | 2675.75 ± 64.72 [a] | 2707.48 ± 85.92 [a] | 11 | 13.68 | 57, 70, 88 |
| Ethyl hexanoate | 2164.56 ± 618.56 [a] | 1536.62 ± 2.19 [a] | 988.41 ± 554.88 [a] | 12 | 15.02 | 88, 115, 144 |
| *p*-Cymene | 525.39 ± 226.28 [a] | 276.66 ± 191.48 [a] | 259.3 ± 124.7 [a] | 13 | 16.85 | 91, 119, 134 |
| (-)-cis-Rose oxide | 17.96 ± 0.19 [a] | 23.95 ± 10.88 [a] | 17.91 ± 0.19 [a] | 14 | 20.56 | 69, 139, 154 |
| **Ethyl octanoate** | 22303.36 ± 3415.44 [a] | 13712.65 ± 1873.27 [ab] | 8322.26 ± 1545.69 [b] | 15 | 24.58 | 43, 71, 86, 116 |
| 1-Octen-3-ol | 4.95 ± 0.55 [a] | 5.02 ± 0.72 [a] | 3.05 ± 0.29 [a] | 16 | 25.25 | 57, 72, 127 |
| Benzaldehyde | 4.19 ± 0.75 [a] | 4.79 ± 0.19 [a] | 4.79 ± 0.32 [a] | 17 | 27.90 | 77, 105, 106 |
| **Ethyl decanoate** | 5967.84 ± 159.97 [a] | 2865.85 ± 352.48 [b] | 1518.58 ± 8.52 | 18 | 33.26 | 43, 71, 86, 116 |
| β-Citronellol | 4.51 ± 0.48 [a] | 5.65 ± 0.49 [a] | 4.72 ± 0.12 [a] | 19 | 38.12 | 123, 137, 152 |
| **Nerol** | 3.52 ± 0.23 [a] | 1.81 ± 0.21 [b] | 1.57 ± 0.06 [b] | 20 | 39.45 | 69, 93, 154 |
| **Damascenone** | 7.29 ± 0.64 [b] | 11.74 ± 1.51 [a] | 8.93 ± 0.25 [b] | 21 | 40.00 | 69, 121, 190 |
| **2-Phenylethyl alcohol** | 3993.5 ± 54.47 [a] | 1287.39 ± 600.72 [b] | 1206.37 ± 3.62 [b] | 22 | 43.17 | 65, 91, 122 |
| β-Ionone | 1.62 ± 0.06 [a] | 2.4 ± 1.01 [a] | 1.78 ± 0.27 [a] | 23 | 44.55 | 135, 177, 192 |
| **Nerolidol** | 1.12 ± 0.01 [b] | 2.12 ± 0.08 [a] | 1.81 ± 0.11 [a] | 24 | 48.16 | 69, 93, 222 |

Statistical differences are expressed as lowercase letters and indicate significant differences in the LSD test (n = 2, $p \leq 0.05$). Means within the row followed by the same letter are not significantly different. [a] RT, retention time. [b] SIM, selected ion monitoring.

### 3.4. Sensory Analysis

During training, the panelists generated 19 terms, 12 for aroma, 4 for taste, and 3 for mouthfeel to describe the 6 wines. A one way-MANOVA for the entire dataset was significant at $p < 0.05$ for the factor wine showing overall differences for the wines. The ANOVA showed that out of the 19 sensory terms, only two aroma terms distinguished the wines significantly at $p < 0.05$, which were 'Sauerkraut aroma' and 'Sulfur (egg) aroma'. Consequently, the least significant difference (LSD) was calculated only for those two terms (Table 4). The letters indicate groupings of wine treatments that were not significantly different). Generally, the recycle semi-batch wines were more intense in 'Sauerkraut' aroma and 'Sulfur (egg)' aroma. However, Control 2 was similar in 'Sauerkraut' aroma to all the treated wines, and the semi-batch 2 wine was comparable to the recycled treatments.

Control 1 had the least 'Sulfur (egg)' aroma and recycle-semi-batch 2 was the most intense in this aroma.

**Table 4.** Significant aroma differences and their Least Significant Differences (LSD). Differences in aroma score greater than LSD indicate a meaningful difference between the samples. Note that letters represent significant groupings based on Fisher's Least Significant Difference (LSD) test $p < 0.05$.

| Wine | Sauerkraut | Sulfur (Egg) |
|---|---|---|
| Control 1 | 0.6 (c) | 0.3 (c) |
| Control 2 | 1.2 (ab) | 0.7 (bc) |
| Semi-Batch 1 | 0.7 (bc) | 0.7 (bc) |
| Semi-Batch 2 | 1.3 (a) | 0.9 (b) |
| Recycle Semi-Batch 1 | 1.3 (a) | 1.1 (b) |
| Recycle Semi-Batch 2 | 1.5 (a) | 1.8 (a) |
| Least Significant Difference | LSD = 0.5 | LSD = 0.6 |

Control 2 is not statistically significantly different from Control 1 for 'Sulfur' aroma and had similar intensity in this aroma to both semi-batch wine replicates and the recycle semi-batch 1 wine.

Overall, we can conclude that the wines were similar in their aroma, taste, and mouthfeel sensory terms; however, the recycle semi-batch wines had significantly more sulfur aromas. In an industrial setting, this may require additional treatment (i.e., carbon filtration, aeration) to ameliorate. Finally, a PCA was performed overlaying the significant differences in chemical and sensory analysis (Figure 3).

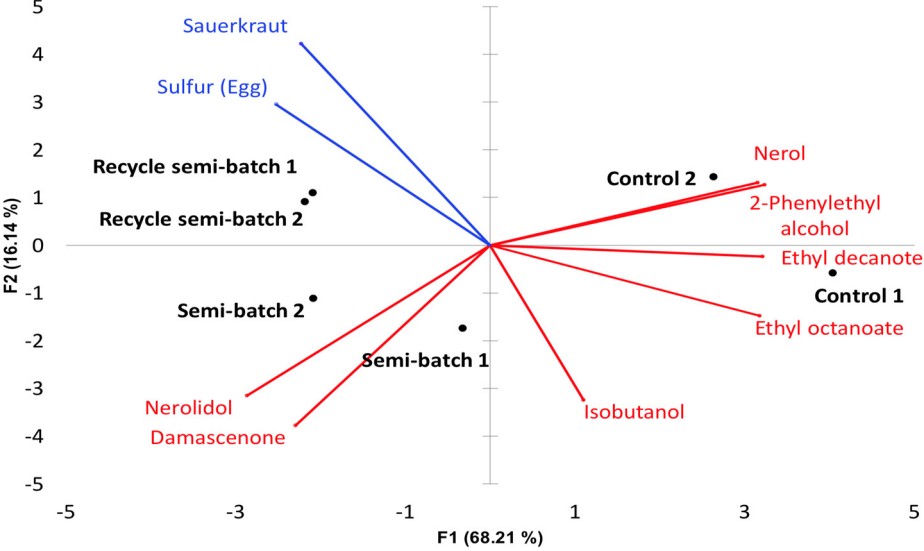

**Figure 3.** Principal Component Analysis (PCA) of seven volatile compounds (red) and two sensory aroma attributes (blue) of six FPI Chardonnay wines (black) that discriminated them significantly at $p < 0.05$. The PCA explained 84.35% of the total variation, 68.21% was explained in the first dimension, and 16.14% in the second dimension.

## 4. Discussion

The results of the pilot and sensory studies indicate that the intensified wine processes can realize substantial cycle time improvements, while using already-available winemaking equipment and risking minimal impact to end wine quality. This represents a major leap forward in wine processing, as it enables wineries to increase their production with minimal capital investment.

Flask studies show that specific YAN uptake drops over the course of fermentation. This is likely due to two phenomena: first, yeast change from primary to secondary metabolism over the course of batch fermentation. Second, the accumulation of ethanol serves to metabolically inhibit yeast cells, reducing the rate of nutrient uptake and division. The application of flask scale data to the pilot trials was successful, as evidenced by the low level of residual YAN in the accelerated fermentations. Ethyl carbamate analysis shows that there is not an associated risk with these large YAN additions.

It is interesting that recycle-semi-batch had less healthy yeast at end of fermentation. Industrial breweries, where cell recycling between fermentations is common, typically use 3–9 "generations" of yeast recycling, due to accumulation of dead cells and strain drift. The semi-batch fermentation had the highest cell viability at the end of fermentation, even higher than the control. This may be due to the fermentation completing earlier, resulting in a shorter exposure time to a high ethanol environment. Additionally, while dry cell weight is highest in recycle-semi-batch fermentation, the cell number is highest in semi-batch. This implies smaller but more abundant cells in the semi-batch fermentation, indicating changes in yeast morphology due to processing conditions.

In calculating YAN additions, fermentation progress was expressed in Brix instead of time. This is a useful independent variable to utilize in wine fermentations, as wine can be fermented over a wide range of temperatures (10–35 °C). Cramer showed that YAN-to-biomass and sugar-to-ethanol yields are not temperature sensitive, so this change of variables is justified and applicable, as the stoichiometries are invariant with temperature. When fitting the flask data, a purely empirical approach yielded an exponential decay fit. This purely empirical fit is convenient for manipulation, but likely does not reflect the underlying metabolic shifts—further investigation into how changing metabolic states impacts nutrient uptake is warranted.

The sensory attributes 'Sauerkraut' and 'Sulfur (egg)' (generally considered negative attributes) are related to sulfur compounds in wines [19–21] that were not measured for this experiment, as they have low sensory detection thresholds. 'Sulfur (egg)' is corresponding to order descriptors of boiled or rotten egg and hydrogen sulfide (H2S) is the compound responsible for them (Siebert et al., 1999). 'Sauerkraut' relates to a cooked cabbage/cabbage odor descriptor. There are several volatile sulfur compounds that could relate to these aromas, such as methanethiol (MeSH) (Solomon et al., 2010), dimethylsulfide (DMS), and dimethyl disulfide DMDS (Goniak and Noble, 1987). It is possible that the rapid fermentation and increased generation number (in the case of the recycle-semi-bath fermentations) contributed to an increase in sulfur aroma compounds. Changes in Ethyl Octanoate are also noticeable and worthy of further investigation. Even still, the detected differences, especially between the control and the semi-batch fermentations, are quite small. While there are differences between the control and the accelerated wines, overall differences are extremely modest, with no difference in 17 out of 19 sensory descriptors.

## 5. Conclusions

In this work, we derived a process to dramatically reduce industrial wine fermentation cycle time, with minimum impact on sensory quality. This study spans from flask-scale fermentation science to pilot-scale fermentation engineering to analytical chemistry to sensory studies. Semi-batch nutrient addition is applicable not only to large commercial wineries, but also to smaller wineries who desire minimal impact on product quality. This technology can increase throughput of existing fermentors with modest investment in laboratory equipment and processing. Recycle semi-batch does require yeast harvesting and seems to increase sulfur notes, but is very applicable to larger scale wineries, especially where blending legs are often carbon filtered or undergo nanofiltration to remove off aromas. This work represents a major step forward in wine fermentation engineering and can be applied to reduce the cost of fermented beverages throughout the world.

**Author Contributions:** K.V.M. (Conceptualization, Formal Analysis, Methodology, Investigation, Writing—original draft, writing—review and editing), E.A. (Investigation), A.A. (Investigation, Writing—original draft), A.C. (Investigation, Formal Analysis, Writing—original draft), R.C.G. (Investigation, Writing—original draft), A.O. (Supervision, Resources), H.H. (Supervision, Resources), and D.E.B. (Conceptualization, Supervision, Resources, Writing—original draft, writing—review and editing). All authors have read and agreed to the published version of the manuscript.

**Funding:** The authors would like to acknowledge funding from American Vineyard Foundation Grant #2209 and the Ernest Gallo Endowed Chair in Viticulture and Enology.

**Institutional Review Board Statement:** This project was approved by the UC Davis Institutional Review Board (IRB) with the IRB project number 1546503-1.

**Informed Consent Statement:** Informed consent was obtained from all subjects involved in the study.

**Data Availability Statement:** All data generated during this study are either included in this article or available from the corresponding author.

**Acknowledgments:** We thank ETS Labs for the donation of Ethyl Carbamate analyses.

**Conflicts of Interest:** The authors declare no conflict of interest.

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
