# Peer review of "Development and Analysis of an Intensified Batch-Fed Wine Fermentation Process"

_fermentation, doi:10.3390/fermentation8060268_

Round 1

Reviewer 1 Report

The study is based on the assumption that wineries can increase the processing capacity and reduce the costs by accelerating the fermentation process. The work hypothesis is that fermentation can be accelerated when YAN is continuously fed to the ferment.

The overall quality of the study is high, the authors chose a highly relevant topic and considered all the relevant details to cover it. The study is interesting from an academic and an engineering perspective.

I suggest to substantiate the work hypothesis by relevant literature. The work hypothesis is that a YAN increase maximizes biomass growth.

The experimental setup and the used analytical methods are appropriate. The authors should provide their data about the biomass.

It is unclear to me how the flask study (= YAN uptake study) and the pilot scale study (fed-batch study) are tied together. Based on the flask study, the YAN uptake decreases with ongoing fermentation. Was this regarded while feeding the ferments in the second study?

The YAN uptake study results in an exponential fit over all treatments which does not seem appropriate. The fast drop at the onset seems to reflect a different metabolic stage of the yeast. The graph is not 100 % clear to me. Is each data point the mean value of n=3? Why is the number of measuring points different from treatment to treatment over the fermentation time?

I disagree to conclude that the sensory impact is minor. The panel was able distinguish the wines by two related off-flavor descriptors which basically show a major quality problem in the fed-batch treatments. To me it seems the other 17 descriptors were not affected much because of a major defects by sulfhydryls.

I also see the massive decreases in ethyl octanoate and decanoate problematic. To me it shows that a lot of open questions need to be answered before the fed-batch technology is ready for the wine industry.

Besides my points, I think this study should be published. The approach bears a huge potential and the overall scientific quality is sound.

Reviewer 2 Report

Only at pag. 3, lines 98-101, please provide reference for OD600 method.

Author Response

We thank Reviewer 2 for their very kind feedback to our paper. We have expanded our method for measuring OD600 in text and added a reference (Sutton 2011)

Reviewer 3 Report

The manuscript presented by Miller and co-authors proposes a new process for managing the nutrition of the yeast during alcoholic fermentation and to reduce wine fermentation cycle time.

Although the study is well executed, there are some aspects that could be detailed or clarified to improve its publication in Fermentation.

Some specific comments about different parts this study are listed below:

1.INTRODUCTION

Line 45. Authors say YAN is typically ranges from 100 to 300 mg/l in grape juice. This is not entirely true since in areas with a warm climate the values ​​are often less than 100 mg/l.

Line 46. The addition of nitrogenous source by the winemaker it's not just only before fermentation. Sometimes it is added once it has started, at the beginning of the exponential phase of yeast growth.

Line 67. Obgonna 1989 is cited. This reference is not detailed in the reference list.

Line 72-73. Authors mention the phenolic extraction as another aspect that can be improved in a fermentation process intensification. This is a fact in red winemaking, but is not the case of this study based in white wine fermentations.

2. MATERIALS AND METHODS

Line 88. …in 500 ml Erlenmeyer flasks… For better understanding authors could add “containing 300 ml of must”.

Line 91. In my opinion high YAN level: 1100 mg/ is extremely high. These values ​​will never be found in a natural must.

Lines 91-92. Low biomass (0.25 g/L) or high biomass (10 g/L). Do you refer to g/L of ADY (Active Dry Yeast) added? g/L of biomass is a bit ambiguous. 10 g/L (= 1000 g/Hl) is an extremely high ADY biomass. The technical data sheets of the main commercial ADY yeast strains recommend between 25-50 g/Hl. The authors would have to justify why they chose such high levels of YAN and biomass.

Line 96. Authors establish the end of fermentations when no change in Brix is recorded over two concurrent measurements. Perhaps it could be better to measure residual sugars by enzymatic techniques.

Lines 101 to 106. Somewhere it would be useful to mention the technique used to measure the YAN, e.g. “YAN analysis were performed by enzymatic/colorimetric determination using a XXXX equipment”.

Line 128. Fermaid K fed twice daily to adjust YAN over the course of fermentation. It would be interesting (here in material and methods or perhaps in results) to indicate how much Fermaid K were added daily during fermentation progress in each case: semi-batch and recycle-semi-batch.

Line 146. Residual sugars were measured… Specify how it was done.

Line 190-191. “This project was approved…” . Why this sentence is mentioned here? I think it is not the section to do it.

Line 208 to 210. Delete this sentence. It is irrelevant to explain this.

3. RESULTS

Line 228. Growth rate (µ_max, 1/h), how is it calculated? Explain here or in material and methods.

Line 229. When Cramer and Coleman are cited, ad the year of publication.

Line 229-230. “As expected, higher YAN results in higher µ_max”. It is not true in the case of high YAN-high biomass. Which is the reason? Explain it.

Line 242-244. In figure caption 1, the sentence “Although all four flask….”, it should be included in the general explanation of section “results”, not in the figure caption.

Line 260 to 264. When authors discuss about residual sugars in the three conditions (referred in table 2): Sentence “The re-cycle semi-batch-fermentation…..”.The values of residual sugars registered are irrelevant. From the point of view of industrial fermentations in wineries, it is considered end of alcoholic fermentation when residual sugars are less than 2 g/l. So, these differences found between batches are not relevant, are not important.

Figure 2. Use the same signal plot for the 3 OD lines: recycle S-B OD is different from the other two.

Line 267 (end) to line 270. Figure caption 2. Sentence “The control slowly reaches a lower ….”. Put it in the comments of results, not in the figure caption.

Table 2. What means End DCW? It should be explained. No reference of DCW is found in the text before.

Line 272 and 273. It should be written in another way what you want to explain is not understandable

Line 286. Authors say that nerolidol is one of the compounds that has higher levels in recycle semi-batch condition. This is not true: higher levels are found in semi-batch condition.

Table 3. Volatile compounds. In general, when results of aromatic compounds are analysed in wines, major volatiles are expressed in mg/L and minor volatiles in µg/L. Here, no units are shown. If an internal standard is used, volatile compounds could have been quantified.

Table 3. For a better understanding of table 3, the aromatic compounds could have been classified by chemical families: alcohols, aldehydes, acids, ethyl esters, terpenes ….

A relationship between compounds and aromatic families (flower, fruit, reduction, lactic, ….) could also have been made. E.g. nerol – flower; isoamyl acetate – fruit (banana),….

Line 296. Table 3 caption. “Green = higher levels ….”. There are no colours in table 3, except compounds in bold. So, it would have to be removed.  

General comments of results

In general, to better understand the behaviour of the fermentation batches and complete this study, it would have been interesting to include the analytical results of the main parameters of the wines obtained in bottle and pilot fermentations: alcoholic strength, total acidity, volatile acidity, pH, glycerol,….

Section 3.3. Wine volatile chemistry analysis should be better discussed.

It would have been interesting to correlate volatile composition with sensory analysis.

Section 3.4. Sauerkraut and sulphur aromas are negative attributes in the aroma profile of wines. It should be mentioned in this section.

4. DISCUSSION

Lines 356 to 358. Authors say that sulphur compounds related to sensory attributes “sauerkraut” and “sulphur-egg” were not measured for this experiment as they have low sensory detection thresholds. However, they have been detected by panellists in the sensory analysis. So, sulphur compounds should be measured.

General comments of discussion

No discussion regarding wine volatile composition have been done.

Sauerkraut and sulphur-egg attributes are negative aromas related to reduction conditions during fermentation, perhaps due to quickly performance of alcoholic fermentations in recycle semi-batch and semi-batch conditions. This could be avoided by micro aeration. This fact and this practice has not been mentioned in the discussion section.

5. CONCLUSIONS

Line 371. The statement that with the new processes proposed there is a minimum impact on sensory quality is not really true. The fact that 13 judges detected with significant differences negative attributes like sauerkraut and sulphur-egg casts doubt on this claim.

REFERENCES

Bibliography should be revised accurately.

Line 406-407. This reference: Arikal et al , 2020 is not cited in the text.

Some references are cited in wrong order:

  • Miller…; 15. Girardello…
  • Siebert…; 19. Solomon…; 20. Goniak…

Line 422. Miller….   2019f. Remove “f”

Author Response

Please see attached responses.

Round 2

Reviewer 3 Report

Why in Table 2 (version 2 of manuscript) Recycle semi Batch trial has been eliminated? Is it a mistake?

CONCLUSIONS. In the first review I said: 

The statement that with the new processes proposed there is a minimum impact on sensory quality is not really true. The fact that 13 judges detected with significant differences negative attributes like sauerkraut and sulphur-egg casts doubt on this claim.

The justification given by the authors is not convincing. In my opinion in  conclusions section, it cannot be stated that this process has a minimal sensory impact because the results do not show that.